# Crack Arrest Effect of FeMnNiSi-Inconel625-Ni60 Laminated Structure Prepared by Laser Cladding Additive Manufacturing

**DOI:** 10.3390/ma18214996

**Published:** 2025-10-31

**Authors:** Lihong Ding, Weining Lei, Jufang Chen

**Affiliations:** 1Engineering Training Center, Jiangsu University of Technology, Changzhou 213001, China; 2Jiangsu Key Laboratory of Advanced Materials Design and Additive Manufacturing, Changzhou 213001, China; leiweining@jsut.edu.cn; 3School of Mechanical Engineering, Jiangsu University of Technology, Changzhou 213001, China; jfchen@jstu.edu.cn

**Keywords:** laser cladding, laminated structure, crack arrest, wear resistance, corrosion resistance

## Abstract

This study addresses the technical challenges of cracking and surface crack initiation in Ni60 alloy cladding layers fabricated by laser cladding additive manufacturing on FeMnNiSi alloys. An innovative FeMnNiSi-Inconel625-Ni60 laminate design was proposed, achieving metallurgical bonding of the dissimilar materials through an Inconel625 transition layer. This effectively addresses the interfacial stress concentration issue caused by differences in thermal expansion coefficients in conventional processes. The results demonstrate that the interfacial microstructure is regulated by synergistic Nb-Mo element segregation, promoting the precipitation of γ″ phase and the formation of a nanoscale Laves phase. This phase not only inhibits carbide aggregation and growth, refining grain size, but also deflects crack propagation paths by pinning dislocations, achieving a dual mechanism of stress reduction and crack arrest. The Ni60 cladding layer in the laminated structure exhibits an average surface microhardness of 641.31 HV_0.3_, 3.88 times that of the substrate (165.22 HV_0.3_), while the Inconel625 base layer shows 340.71 HV_0.3_, 2.06 times the substrate’s value. Wear testing reveals the laminated cladding layer has a wear volume of 0.086 mm^3^ (0.243 mm^3^ less than the substrate’s 0.329 mm^3^) and a wear rate of 0.86 × 10^−2^ mm^3^/(N·m), 73.86% lower than the substrate’s 3.29 × 10^−2^ mm^3^/(N·m), indicating superior wear resistance. The electrochemical test results show that under the same corrosion conditions, the self-corrosion potential and polarization resistance of the FeMnNiSi-Inconel625-Ni60 cladding layer are significantly higher than those of the substrate, while the corrosion current density is significantly lower than that of the substrate. The frequency stability region at the highest impedance modulus |Z| is wider than that of the substrate, and the corrosion rate is 71.86% slower than that of the substrate, demonstrating excellent wear resistance. This study not only reveals the mechanism by which Laves phases improve interfacial properties through microstructural regulation but also provides a scalable interface design strategy for heterogeneous material additive manufacturing, which has important engineering value in promoting the application of laser cladding technology in the field of high-end equipment repair.

## 1. Introduction

FeMnNiSi mold steel, as a typical low-cost medium-carbon steel, has a carbon content of about 0.45%. Due to its excellent processing performance and good comprehensive mechanical properties, it is widely used in mold manufacturing, mechanical transmission components, and fixtures [1]. However, this material has significant deficiencies in hardness, wear resistance, and corrosion resistance. Especially under high-load and high-wear conditions, it is very prone to wear failure and plastic deformation.

Laser cladding technology uses high-energy laser beams to achieve surface modification of materials and has outstanding performance in improving tribological properties. Compared with the two-phase process (nitriding), which is only suitable for surface hardening, laser cladding can directly repair worn parts. Its core advantage lies not only in the depth of the nitriding layer but also in its ability to simultaneously achieve alloying, cladding, and strengthening, and it is suitable for complex geometric shapes and various material systems. The non-equilibrium structure formed through the rapid cooling of this process makes the deposited layer usually present a fine-grained structure, with low porosity and metallurgical bonding characteristics. Its heat-affected zone (HAZ) is narrow, and the energy input can be precisely controlled, thereby reducing the thermal deformation of the substrate [2,3]. Due to its unique advantages, laser cladding technology has been widely used in material surface strengthening in recent years. However, its fatigue life improvement effect under compressive stress is limited and is greatly affected by material and process parameters [4].

In response to this problem, domestic and foreign scholars have conducted in-depth research on improving the surface properties of steel using laser cladding technology. Li T et al. [5] conducted multi-objective optimization of the process parameters of laser cladding Ni60 alloy powder on 45 steel. The optimized process parameters can obtain a cladding structure with refined grains and improve the surface properties of 45 steel. Luo Xixi et al. [6] studied the preparation of an Al-Cr composite strengthening layer on the surface of 45 steel and verified that a composite passivation film was formed on the surface of the Al-Cr coating with excellent corrosion resistance. Liu Lilan et al. [7] proposed a multivariate regression prediction method for optimization targets. The results showed that in the laser cladding process, increasing the laser power and reducing the powder feeding rate and scanning speed can reduce the crack density and improve the cladding performance. Cao Jinlong et al. [8] conducted research on the Ni60-TiC gradient layer structure and studied the cladding of TiC coating on the surface of 45 steel using a CO_2_ constant current laser. Wang Sheng et al. [9] prepared Fe35 on the surface of 45 steel material. An alloy layer, under laser cladding parameters of a laser power of 2100 W, scanning speed of 5 mm/s, and powder feeding speed of 15 g/min, can obtain fine and uniform grains, effectively reducing brittle carbides. Wang Yuhao et al. [10] used electromagnetic field-assisted laser cladding technology to prepare a dense Ni60 coating on the surface of 45 steel. As a result, the stirring effect of the surface electromagnetic field on the molten pool will accelerate the heat and mass transfer of the cladding layer, thereby reducing the generation of substances such as CrB and Ni_2_B and enhancing the surface properties of 45 steel.

Although the above researchers have made some progress, there are still some shortcomings in the following aspects:Most of the existing studies prepare a single coating on the substrate surface, and there is a lack of research on laminated structures.Most of the existing studies prepare specific alloy materials Ni60 or coating Fe_35_A on the substrate surface and optimize the process parameters, but lack the universal verification of the material system.Most of the existing research aims to refine grains, reduce cracks, and enhance the fusion between substrate and cladding by optimizing process parameters and increasing surface electromagnetic fields, but there is no targeted related research.

Ni60 has a wide range of application prospects in the field of surface engineering due to its unique performance advantages. Not only does this alloy have excellent corrosion resistance, high hardness, and good wear resistance, but also in injection mold applications, the Ni60 cladding layer can extend mold life by 3–5 times and reduce maintenance costs by more than 40% [11,12]. The advantages of Ni60 alloy effectively make up for the shortcomings of FeMnNiSi steel performance, making it an ideal material for FeMnNiSi steel laser cladding [13]. However, the Ni60 cladding layer is prone to generate high residual stress during rapid solidification, resulting in a significant cracking tendency, especially the initiation and expansion of microcracks, which seriously restricts its service reliability in harsh environments [14,15,16,17]. The surface modification of the Ni60 cladding layer by laser cladding on FeMnNiSi steel faces two key technical difficulties in the preparation process: one is the residual stress problem caused by the difference in the thermal expansion coefficient between Ni60 and the substrate, which can easily cause longitudinal cracks [18]; the other is the formation of brittle carbide Cr_7_C_3_ in the interface area, which reduces the bonding strength by 30–40% [19].

This study addresses the technical challenge of cracking by laser-cladding Ni60 alloy on FeMnNiSi alloy. Through the proposal of an innovative design using an Inconel625 cladding layer as an intermediate transition layer, the goal is to construct an FeMnNiSi-Inconel625-Ni60 laminate structure, mitigating the difference in thermal expansion coefficients between FeMnNiSi and Ni60 while simultaneously achieving a metallurgical interlock between the FeMnNiSi matrix and the Ni60 functional layer. Not only does this research provide a theoretical basis for the additive manufacturing of heterogeneous materials, but its technical approach can also be directly applied to the surface modification of key components in aircraft engines, marine equipment, and other applications requiring synergistic wear and corrosion resistance, possessing significant engineering application value.

## 2. Experimental Materials and Methods

### 2.1. Test Materials

The substrate material employed in this study was FeMnNiSi die steel with dimensions of 100 mm × 100 mm × 10 mm. Two nickel-based alloy powders were selected for laser cladding: Ni60 (supplied by AVIC Zhongmai Metal Materials Research Institute, Beijing, China) and Inconel625 (supplied by Höganäs AB, Höganäs, Sweden). To ensure suitability for laser cladding processing, the powders were sieved to a particle size range of 45~105 μm, with a purity exceeding 99.9% and spheroidal morphology. Prior to laser cladding, both Ni60 and Inconel625 powders were subjected to a drying process at 200 °C for 2 h in an FZG-8 vacuum drying oven to eliminate moisture and enhance experimental precision. Table 1 presents the chemical composition and elemental content of the substrate, Ni60, and Inconel625.

### 2.2. Laser Cladding Process

The laser cladding test is a laser cladding forming system composed of a 4 kW IPG laser, KR60HA KUKA robot (Tennessee Industrial Electronics, La Vergne, TN, USA), GTV-adjustable dual-material bin, airborne powder feeder, and worktable. The YLS-4000 fiber laser (IPG Photonics, Oxford, MA, USA) was used for single-channel and single-layer laser cladding tests with four synchronous coaxial powder feeders. The laser power was 1.5 kW, the spot diameter was 3 mm, the scanning speed was 6 mm/s, and the powder feeding rate was 10 g/min. During the laser cladding process, high-purity inert argon gas was used for protection, and the gas flow rate was set to 7 L/min. Two different cladding layers were prepared in the experiment, namely FeMnNiSi-Ni60 single cladding layer and FeMnNiSi-Inconel625-Ni60 laminated structure cladding layer.

### 2.3. Preparation and Characterization of Test Samples

Three sample geometries (10 × 10 × 10 mm^3^, 20 × 10 × 10 mm^3^, 10 × 10 × 2 mm^3^) were sectioned perpendicular to the laser scan direction using an SG3000 electric discharge wire cutter (Suzhou Sanguang Science & Technology Co., Suzhou, China). For microstructural analysis, the 10 × 10 × 10 mm^3^ samples underwent cross-sectional grinding/polishing, followed by ultrasonic cleaning in anhydrous ethanol and chemical etching with a mixed solution of 20 mL HNO_3_ and 20 mL HCl. Microhardness measurements (300 gf load, 10 s dwell time) were performed on the cladding layer using an HXD-1000TMS tester (SDL Atlas, Singapore), with three readings averaged per horizontal line.

Wear resistance was evaluated on 20 × 10 × 10 mm^3^ samples polished to 2000 grit. Pre- and post-test mass measurements (0.1 mg precision balance) were conducted after 30 min sliding against G5-G100 Si_3_N_4_ ceramic balls (6 mm diameter, 20 N load, 5 mm stroke). Three-dimensional wear scar profiles were quantified via NANOPS50 profilometry (SCANTECH, Hangzhou, China).

Corrosion behavior was assessed via potentiodynamic polarization tests (1 cm^2^ exposed area) using a CHI600E workstation (Shanghai Chenhua, Shanghai, China) on polished 10 × 10 × 10 mm^3^ samples. Phase analysis of the 10 × 10 × 2 mm^3^ samples employed XRD (HD XpertPRO (ZEISS, Oberkochen, Germany)), while cross-sectional microstructural examination utilized SEM (SIGMA500 (ZEISS, Oberkochen, Germany)) with EDS (INCA OXFORD (Oxford Instruments, Abingdon, UK)) for elemental mapping.

## 3. Test Results

### 3.1. Interface Micromorphology

The micromorphology of the interface region between the FeMnNiSi-Ni60 single cladding layer and the FeMnNiSi-Inconel625-Ni60 laminated structure was observed under a scanning electron microscope. Figure 1(a-1,a-2) show the micromorphology of the interface region of the single cladding layer; Figure 1(b-1) shows the micromorphology of the FeMnNiSi-Inconel625 laminated structure interface, and Figure 1(b-2) shows the micromorphology of the Inconel625-Ni60 laminated structure interface. Figure 1(c-1) shows the micromorphology of the Inconel625 transition layer of the laminated structure, and Figure 1(c-2) shows the micromorphology of the Ni60 laminated structure cladding layer.

As shown in Figure 1(a-1), there is a black gap defect parallel to the substrate in the fusion zone. This is a non-fusion phenomenon caused by insufficient fluidity of the molten pool. Its outline is clear and linearly distributed, which conforms to the morphological characteristics of typical non-fusion defects. As shown in Figure 1(a-2), there is a large difference in the thermal expansion coefficient between the substrate FeMnNiSi and the Ni60 cladding layer. The difference in contraction during cooling produces high tensile stress. Ni60 itself is brittle and has weak tensile stress resistance, which leads to cracks initiating in the fusion zone and extending to the cladding layer area, presenting a transgranular fracture micromorphology.

In Figure 1(b-1), no obvious cladding defects are observed at the interface of the laminated structure FeMnNiSi-Inconel625. Transition layer Inconel625 has excellent plasticity and a thermal expansion coefficient similar to matrix FeMnNiSi, effectively alleviating thermal stress concentration. As shown in Figure 1(b-2), there are also no obvious cladding defects at the interface of the laminated structure Inconel625-Ni60. The cladding layer Inconel625 has a face-centered cubic structure and solid solution-strengthening effect. Its high toughness forms a gradient transition with Ni60, inhibiting crack propagation [20].

In Figure 1(c-1), at a magnification of 20,000 times, the micromorphology of the Inconel625 transition layer of the laminated structure reveals the precipitation of multiple white bright spots in the interdendritic gaps. Analysis of the percentage of alloying elements at each point reveals that this phase is a nanoscale Laves phase, which can reduce the segregation of harmful elements (such as Cr and Mo) at grain boundaries, reducing grain boundary brittleness. At the same time, pinning the grain boundaries inhibits excessive grain growth, achieving grain refinement and effectively suppressing the generation of cladding defects. A γ′ phase is also generated in the transition layer region, forming a coherent strain field with the matrix. The mismatch is only 0.08%, significantly improving the yield strength of the alloy. The uniform precipitation of the γ′ phase can greatly increase the room temperature tensile strength, which also reflects the crack arrest effect of the FeMnNiSi-Inconel625-Ni60 laminated structure. As shown in Figure 1(c-2), at a magnification of 5000 times, a γ-Ni solid solution phase is observed in the Ni60 laminated surface cladding layer. This phase provides plastic deformation capability, preventing the spalling of the cladding layer due to the brittleness of carbides. Furthermore, the γ-Ni matrix absorbs impact energy through plastic deformation, slowing crack propagation. Furthermore, a high-hardness and wear-resistant (Mo,Nb)C or (Mo,Cr)C solid solution-strengthening phase is generated in the surface cladding layer [21]. The coherent interface (stellate precipitation structure) with the γ-Ni matrix reduces stress concentration and improves the bonding strength between the cladding layer and the substrate.

In summary, from the microstructural observation, it was found that the laminated structure FeMnNiSi-Inconel625-Ni60 effectively inhibited the initiation and propagation of cracks through the dual design of transition layer strengthening and cladding layer toughening, providing a stable foundation for matrix surface strengthening.

### 3.2. Interface Element Analysis

To investigate the crack-arresting mechanism of the FeMnNiSi-Inconel625-Ni60 laminated structure, we performed scanning electron microscopy (SEM) coupled with energy-dispersive X-ray spectroscopy (EDS) using an INCA OXFORD system. This enabled simultaneous microstructural characterization and elemental mapping at the substrate-cladding interface. Figure 2 shows the microstructure and element distribution curve of the FeMnNiSi-Ni60 single cladding layer interface; Figure 3 shows the microstructure and element distribution curve of the FeMnNiSi-Inconel625 interface in the FeMnNiSi-Inconel625-Ni60 laminated structure; and Figure 4 shows the microstructure and element distribution of the Inconel625-Ni60 interface in the FeMnNiSi-Inconel625-Ni60 laminated structure.

As shown in Figure 2, the black phase area of the online scanning shows the phenomenon of multiple-Cr-element aggregation and a significant reduction in Ni element. Cr-rich areas are prone to forming hard and brittle BCC phases, such as Cr_23_C_6_ carbides, which can lead to cladding defects. While Cr elements are gathering, the Ni element content is significantly reduced. The Ni element reduction area is mainly composed of FCC phases. The difference in thermal expansion coefficients between the BCC phase and the FCC phase leads to interfacial stress concentration, thereby increasing the risk of crack generation. On the other hand, Cr and Ni elements show opposite trends of change. The anti-correlated distribution of Cr-Ni promotes the precipitation of brittle phases such as (Cr, Fe)_23_C_6_ at the grain boundary. The interface between these hard phases and the substrate often becomes the area where cracks preferentially extend [22]. It is because of the above situation that cracks and obvious cladding defects were found in the fusion zone in the microstructure observation of a single cladding layer.

Figure 3 shows the elemental analysis of the FeMnNiSi-Inconel625 interface of the laminated structure. It can be seen from the observation that in the line scanning area of the Inconel625 cladding layer, Nb and Mo elements are concentrated in the white phase, while Fe, Cr, and Ni elements are reduced. The enrichment of Nb and Mo elements will promote the precipitation of γ″ phase Ni_3_Nb. Its coherent strain with the matrix can absorb residual stress and reduce the local stress concentration coefficient. In addition, the synergistic segregation of Nb-Mo can form a nanoscale Laves phase, which can achieve stress redistribution by pinning dislocation movement. The study of Rao G G and other phase field methods shows that Nb Mo co-segregation can increase the energy barrier of dislocation motion by 0.8 eV, which is consistent with the stress redistribution phenomenon achieved by pinning dislocation motion in this study [23]. The segregation of Nb can stabilize grain boundary carbides and reduce the initiation of microcracks caused by grain boundary sliding at high temperatures. The aggregation of Mo can increase the local yield strength through solid solution strengthening, so that crack propagation requires higher energy consumption, thereby reducing the occurrence of cracks [24]. The reduction in Fe and Cr can inhibit the formation of brittle σ phase Cr-Fe and avoid stress mutation caused by hard and brittle phase. The moderate reduction in Ni can reduce the difference in the thermal expansion coefficient between the Inconel625 cladding layer and FeMnNiSi substrate layer, as well as reduce interface thermal stress, and the reduction in Cr can avoid the continuous precipitation of carbides at grain boundaries and improve the grain boundary bonding state [25]. Through the above element analysis, we found that for the laminated structure with Inconel625 as the intermediate laminate, its bonding interface with the substrate presents a dual mechanism of reducing stress concentration and inhibiting cracks.

Figure 4 shows the elemental analysis of the Inconel625-Ni60 interface of the laminated structure. Via observation, it can be found that there are many fine white phases distributed throughout the interface. The positions of these phases all show the aggregation of Nb or Mo elements. In particular, there are two areas in the fusion line area where Nb and Mo elements are aggregated, while Cr and Ni elements are reduced. This phenomenon shows that Nb-Mo synergistically segregates in the fusion line area to form a nanoscale Laves phase, which can not only inhibit the formation and aggregation growth of carbides, thereby achieving the grain refinement effect, but also form pinning dislocations, causing the crack propagation path to deflect and reducing crack generation [26].

### 3.3. Phase Composition Analysis

To further verify the crack arrest effect of the FeMnNiSi-Inconel625-Ni60 laminated structure, diffraction pattern analysis of the laminated structure was carried out. The black broken line in Figure 5 is the XRD diffraction pattern of the Inconel625 cladding layer, and the red broken line is the XRD diffraction pattern of the Ni60 cladding layer. The two samples are the transition layer and the surface cladding layer of the laminated structure cladding layer, respectively. The characteristic peak positions of the FCC phase of the Inconel625 cladding layer are at 44.2°, 51.5° and 75.7°, the diffraction peaks of the Laves phase are at 38.3°and 56.2°, and the diffraction peaks of the γ″ phase are at 42.1° and 46.1°. The characteristic peak positions of the FCC phase of the Ni60 cladding layer are at 44.8°, 52°, and 76.2°.

Because this area is a surface cladding layer of a laminated structure, Nb and Mo elements are segregated from the Inconel625 cladding layer to this area, so the diffraction peak here may also correspond to the Laves phase or γ″ phase.

In comparing the specific data of the crystal plane diffraction peaks of the two cladding layers, it was found that the diffraction peaks of the FCC phase of the Inconel625 cladding layer shifted to the left. This is because the high content of Nb and Mo and other large-atomic-radius elements in Inconel625 replace the Ni lattice, resulting in the expansion of the γ-Ni lattice. According to the Bragg equation, 2d·sinθ = nλ, the increase in the interplanar spacing d will lead to a decrease in the diffraction angle θ, which is manifested as a left shift in the peak position. The presence of FCC phase in both cladding layers indicates that the cladding layers have good high-temperature stability, and the Cr element in the cladding layer can promote the formation of the passivation film and improve the corrosion resistance of the cladding layer in 3.5% NaCl solution [25]. The Laves phase acts as a pinning phase at the grain boundary to inhibit grain boundary sliding and maintain the stability of the structure under high-temperature conditions of laser cladding. Moreover, the preferential precipitation of the Laves phase can consume Nb, reduce the precipitation amount of NbC carbides, and reduce the peak strength of carbides [26]. The γ″ phase pins the grain boundary to hinder the continuous network precipitation of carbides, achieving a grain refinement effect and dominating the low-temperature strengthening. The synergistic effect of the two structures can improve the performance of the cladding layer and effectively inhibit the generation of cracks.

### 3.4. Microhardness

Regarding the microhardness measurement of the FeMnNiSi-Inconel625-Ni60 laminated structure cross-section, microhardness measurements were performed along a horizontal cross-section of the cladding layer, with test points spaced at 200 μm intervals from the cladding surface to the substrate. The average microhardness values (HV_0.3_) reveal a distinct gradient: the substrate (FeMnNiSi) exhibits a hardness of 165.22 HV_0.3_, while the intermediate Inconel 625 cladding layer reaches 340.71 HV_0.3_ (2.06× the substrate). The surface Ni60 layer achieves the highest hardness at 641.31 HV_0.3_ (3.88× the substrate), as shown in Figure 6. This progressive increase in hardness—from substrate to surface—suggests that the Inconel625 layer acts as an effective buffer, mitigating stress concentrations between the softer substrate and the harder Ni60 surface layer. Such a graded hardness profile is likely to enhance the mechanical integrity and wear resistance of the laminated structure. Sainath Krishna Mani Iyer found in the study of SS316L/Inconel625-based composite functional gradient coatings that the transition of the intermediate layer reduces interfacial stress concentration by more than 40%, providing theoretical support for this study [27]. During data recording, we also discovered that the hardness values of one area between the different cladding layers differed, as did the hardness of the two layers. This phenomenon is primarily due to the significant differences in hardness between the different alloy materials. While the cladding retains its original properties, the laser heat input causes elemental migration. During the laser cladding process, some elements also generate new species, all of which lead to changes in hardness. Overall hardness analysis shows that the cladding layer in the FeMnNiSi-Inconel625-Ni60 stack is significantly harder than the substrate, achieving a surface strengthening effect.

### 3.5. Friction and Wear Properties

As shown in Figure 7a, the friction coefficient of both the substrate and the laminated cladding layer exhibits an initial transient period before stabilizing after approximately 5 min, indicating the transition into steady-state wear. The stabilized average friction coefficient (Figure 7b) reveals a 17.95% increase for the laminated structure (0.552) compared to the substrate (0.543). According to Archard’s wear theory (Equation (1)), wear volume (V) is directly proportional to the sliding distance (S) and normal load (N), while inversely proportional to material hardness (H) [28].(1)V=kNS/H

Under identical testing conditions (20 N load, 5 mm sliding distance), the wear volume of the substrate (0.329 mm^3^) significantly exceeds that of the laminated cladding layer (0.086 mm^3^)—a reduction of 0.243 mm^3^ (Figure 8b). The specific parameters are shown in Table 2. This pronounced improvement in wear resistance can be attributed to the graded hardness profile of the laminated structure, where the Inconel625 intermediate layer and Ni60 surface layer collectively enhance durability while maintaining a modest increase in friction coefficient.

To further evaluate the friction and wear, the wear scars of the cladding layer and the substrate were extracted and characterized using 3D contours. Figure 8a shows the 3D contour of the wear scar of the substrate, and Figure 8b shows the 3D contour of the wear scar of the FeMnNiSi-Inconel625-Ni60 laminated cladding layer. The two figures show that the wear scar of the laminated cladding layer is shallow and narrow, while the wear scar of the substrate is deep and wide, visually verifying that the wear resistance of the FeMnNiSi-Inconel625-Ni60 laminated cladding layer is significantly higher than that of the substrate.

The three-dimensional contour morphology of the scratch was characterized, yielding the comprehensive dataset presented in Table 3. In order to further verify the advantages of the laminated structure, the wear rate was calculated by substituting Formula (2) into the data obtained by scanning the three-dimensional profile [29]. The specific formula is as follows:(2)$$W\;=\;\frac{V}{\mathrm{LF}}$$

In Formula (2), *V* is the wear volume; *L* is the wear distance; *F* is the wear load; and *W* is the wear rate.

Wear scar morphology analysis (Table 3) reveals that the FeMnNiSi-Inconel625-Ni60 laminated cladding layer exhibits superior wear resistance compared to the substrate. Specifically, the maximum wear depth (16.5 μm vs. 18.6 μm) and cross-sectional area (3883 μm^2^ vs. 7806 μm^2^) are significantly reduced, leading to a 73.86% decrease in wear rate (0.86 × 10^−2^ mm^3^/(N·m) vs. 3.29 × 10^−2^ mm^3^/(N·m)). These findings demonstrate that the laminated cladding structure effectively enhances the wear resistance of FeMnNiSi steel.

Scanning electron microscopy (SEM) examination of the wear scar morphology (Figure 9) reveals distinct wear mechanisms between the substrate and the laminated cladding layer. The substrate (Figure 9a) exhibits severe adhesive wear characteristics, including thick material peeling layers, deep grooves, and extensive material transfer, accompanied by abrasive wear due to large grinding debris accumulation. In contrast, the laminated cladding layer (Figure 9b) demonstrates primarily abrasive wear behavior, with fine grinding debris particles generated by the strengthening phase under grinding force, forming shallow grooves alongside the cutting action of silicon nitrideceramic balls. Minor adhesive wear features are observed, but no large-scale material peeling occurs [30].

The superior wear resistance of the laminated cladding layer is attributed to two synergistic effects: (1) solid solution strengthening and grain refinement, which enhance hardness and deformation resistance; and (2) a strengthening phase that impedes dislocation motion, further suppressing wear. In comparison, the substrate’s lower hardness renders it susceptible to failure under combined normal and shear stresses during abrasive contact [31].

### 3.6. Corrosion Resistance

Figure 10 presents the Tafel polarization curves of the FeMnNiSi-Inconel625-Ni60 laminated cladding layer and the substrate in 3.5% NaCl solution. A marked difference in electrochemical behavior is evident: the laminated cladding layer displays a well-defined smooth-curve region during corrosion, whereas the substrate lacks such a feature entirely. This smooth region represents the passivation region, indicating the good stability of the passivation layer. This demonstrates that the laminated cladding layer exhibits superior corrosion resistance compared to the substrate. Polarization resistance (Rp) directly reflects a material’s ability to resist corrosion. The greater the polarization resistance, the greater the corrosion resistance. It is inversely proportional to the corrosion current density, and the proportional value can be obtained from the slope of the polarization curve. In electrochemical experiments, the polarization curve data measured in CS Studio software (SP5) was imported into CS (Version: V2.1.586) Analysis software for analysis. CS Analysis software automatically analyzes the polarization curve and calculates the polarization resistance Rp. The data show that the Rp values of the substrate and the laminated cladding are 2426.9 Ω and 8623.6 Ω, respectively. The laminated cladding exhibits markedly improved corrosion resistance relative to the substrate, featuring a 71.86% reduction in corrosion rate and elevated polarization resistance. Electrochemical analysis indicates that the cladding’s corrosion potential (Ecorr) (−0.780 V) is more positive than that of the substrate (−0.844 V), while its corrosion current density (icorr) (8.32 × 10^−6^ A/cm^2^) is approximately one order of magnitude lower than the substrate’s value (6.546 × 10^−5^ A/cm^2^). These findings collectively demonstrate that the laminated structure achieves slower surface degradation, diminished corrosion propensity, and enhanced protective performance under uniform environmental conditions [32].

To elucidate the electrochemical corrosion behavior of the FeMnNiSi-Inconel625-Ni60 laminated cladding layer and its substrate, electrochemical impedance spectroscopy (EIS) analysis was conducted, as illustrated in Figure 11. Figure 11a presents the Nyquist plots for both the laminated cladding and the substrate, while Figure 11b shows the corresponding Bode diagrams.

In Figure 11a, the laminated cladding exhibits a substantially larger capacitive arc radius than the substrate. A larger capacitive arc radius correlates with higher resistance to electrode reactions, slower corrosion rates, and superior corrosion resistance [33]. Figure 11b further reveals that the impedance modulus |Z| of the laminated cladding is significantly higher than that of the substrate. Notably, at the peak |Z| frequency [34], the laminated cladding demonstrates a broad frequency stability region, whereas this region is virtually absent in the substrate. These observations collectively confirm the enhanced corrosion resistance of the FeMnNiSi-Inconel625-Ni60 laminated structure.

Figure 12 shows the microscopic morphology of the substrate and cladding surface after electrochemical corrosion. As shown in Figure 12a, the corroded surface of the substrate has obvious pitting corrosion pits, and the substrate test surface is corroded as a whole, with basically no intact areas. Compared with Figure 12b, there are areas of uncorroded cladding in the laminated structure, and a small amount of clustered pitting corrosion clusters is present within the corroded grain boundaries. However, the corrosion marks are relatively shallow, indicating that the laminated structure cladding has better corrosion resistance. Therefore, in terms of corrosion resistance, the laminated structure cladding has better corrosion resistance than the substrate.

## 4. Conclusions

This study successfully solved the cracking problem of Ni60 cladding layer on FeMnNiSi substrate through the design of a FeMnNiSi-Inconel625-Ni60 laminate structure, as well as revealed the intrinsic mechanism of interface microstructure regulation and performance enhancement and provided a theoretical basis for the additive manufacturing of heterogeneous materials. It also provided a basis for the surface modification of key components in industrial and mining applications such as aviation and marine equipment. The main conclusions are as follows:(1)By observing the micromorphology of the interface area between the Ni60 single cladding layer and the FeMnNiSi-Inconel625-Ni60 laminated structure, we found unfused defects and obvious cracks in the fusion zone of the single cladding layer, while there were no obvious cladding defects in the two cladding layer areas of the laminated structure.(2)Through EBSD micromorphology, interface elements, and phase composition analysis, it can be seen that the Cr element aggregates at the interface of Ni60 single coating and the Ni element decreases, thus forming a hard brittle phase and FCC phase. The difference in the thermal expansion coefficients of the two phases leads to interface stress concentration, which increases the risk of crack generation; the enrichment of Nb and Mo elements at the interface of the stacked structure FeMnNiSi-Inconel625 promotes the precipitation of γ″ phase and forms a nanoscale Laves phase under the synergistic segregation of Nb-Mo, which refines the grains while reducing local stress concentration; two Nb-Mo synergistic segregation phenomena appear in the fusion line area of the stacked structure Inconel625-Ni60 interface, which can not only inhibit the formation and aggregation of carbides to achieve the effect of grain refinement but also form pinning dislocations, deflect the crack propagation path, and reduce crack generation. The FeMnNiSi-Inconel625-Ni60 stacked structure presents a dual mechanism of reducing stress concentration and inhibiting cracks.(3)The hardness test results show that the average microhardness of the Ni60 surface cladding layer of the laminated structure is 641.31 HV_0.3_, which is 3.88 times the average microhardness of the substrate (165.22 HV_0.3_). The average microhardness of the base layer Inconel625 is 340.71 HV_0.3_, which is 2.06 times the microhardness of the substrate. The wear resistance test results show that the wear volume of the laminated cladding layer is 0.086 mm^3^, which is 0.243 mm^3^ less than the wear volume of the substrate 0.329 mm^3^; the wear rate of the laminated cladding layer is 0.86 × 10^−2^ mm^3^/(N·m), which is 73.86% lower than the wear rate of the substrate 3.29 × 10^−2^ mm^3^/(N·m); the substrate wear scar surface has a large area of adhesion and material peeling, resulting in severe wear, while the surface wear of the laminated cladding layer is significantly weakened, mainly due to abrasive wear, and has better wear resistance.(4)The electrochemical corrosion test results indicate that under identical environmental conditions, the laminated structure cladding layer exhibits superior corrosion resistance compared to the substrate. Specifically, the cladding layer demonstrates a higher self-corrosion potential (−0.780 V vs. −0.844 V for the substrate), a lower corrosion current density (8.32 × 10^−6^ A/cm^2^ vs. 6.546 × 10^−5^ A/cm^2^), and a significantly higher polarization resistance (8623.6 Ω vs. 2426.9 Ω). Additionally, the frequency-stable region width at the highest impedance modulus |Z| is notably wider for the cladding layer, and its corrosion rate is 71.86% slower than that of the substrate. These findings confirm that the laminated structure has a slow corrosion rate, minimal corrosion tendency, and excellent overall corrosion resistance.

## Figures and Tables

**Figure 1 materials-18-04996-f001:**
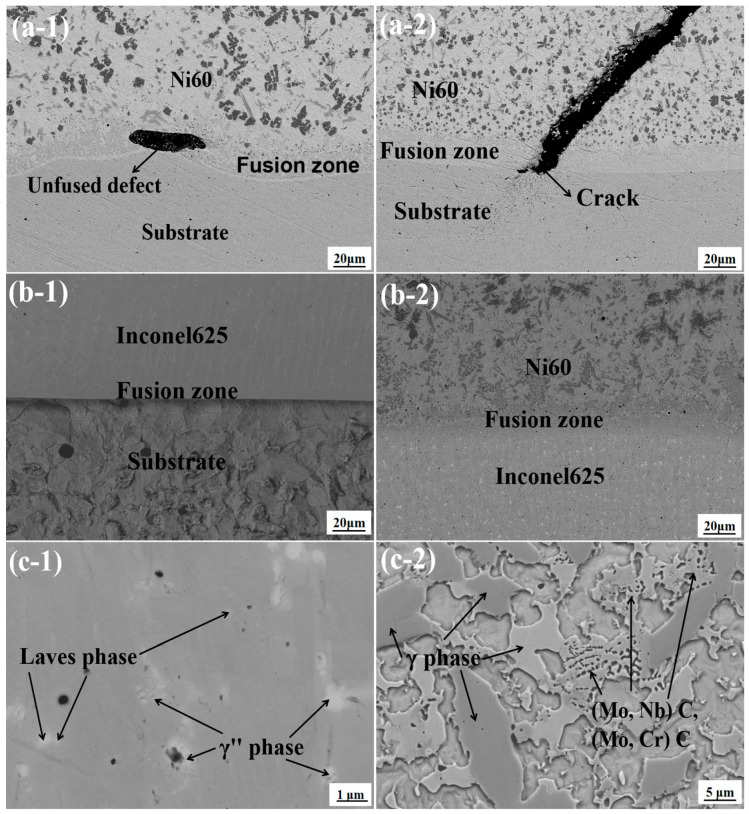
Interface micromorphology. (**a-1**) Unfused defect at the interface of a single cladding layer. (**a-2**) Crack at the interface of a single cladding layer. (**b-1**) Micromorphology of the FeMnNiSi-Inconel625 interface of the laminated structure. (**b-2**) Micromorphology of the Inconel625-Ni60 interface of the laminated structure. (**c-1**) micromorphology of Inconel625 transition layer in laminated structure (**c-2**) micromorphology of Ni60 cladding layer in laminated structure.

**Figure 2 materials-18-04996-f002:**
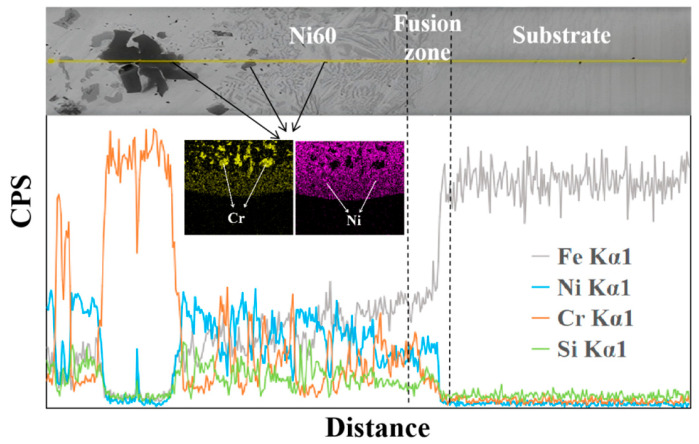
Element analysis of FeMnNiSi-Ni60 interface.

**Figure 3 materials-18-04996-f003:**
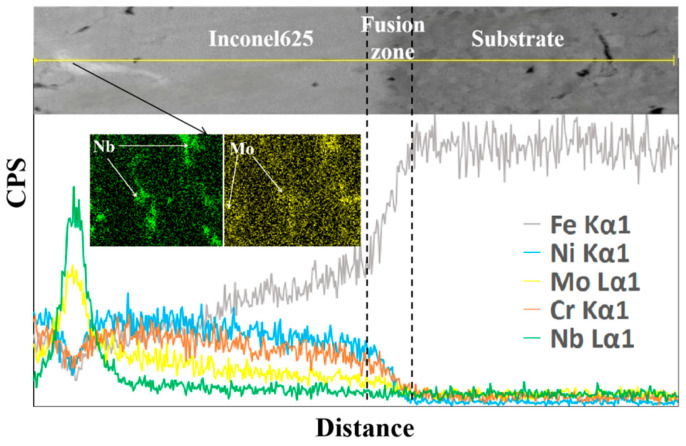
Element analysis of FeMnNiSi-Inconel625 interface.

**Figure 4 materials-18-04996-f004:**
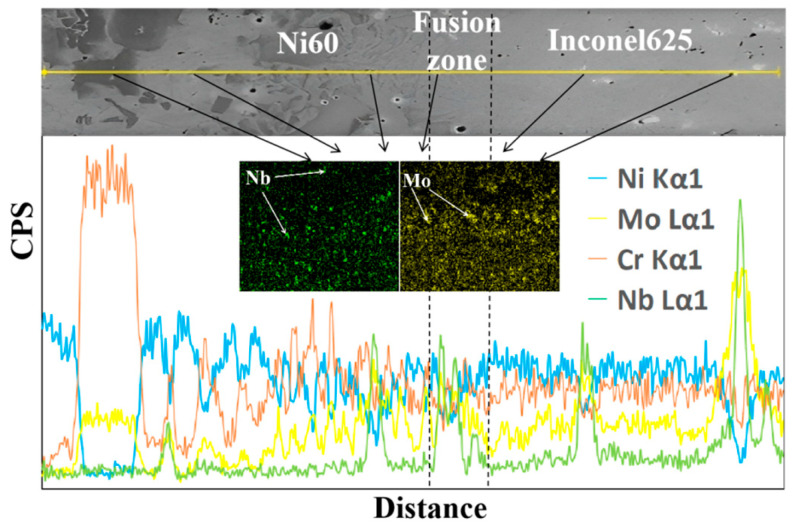
SEM and EDS of Inconel625-Ni60 interface.

**Figure 5 materials-18-04996-f005:**
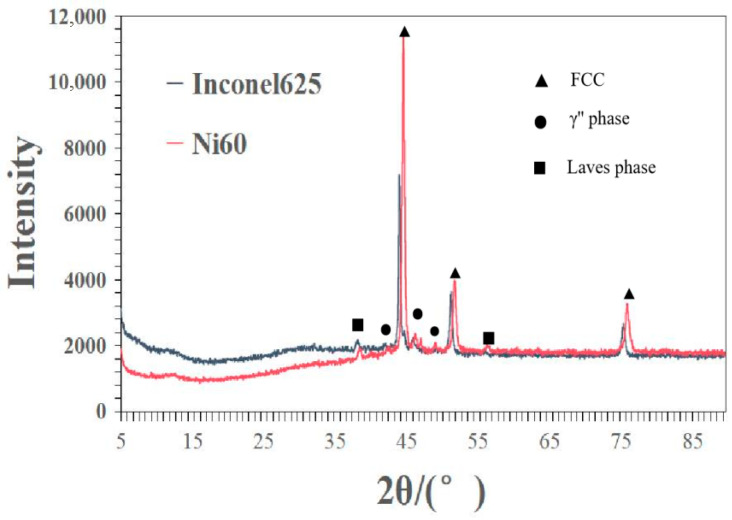
XRD diffraction pattern of the phase composition of the cladding layer.

**Figure 6 materials-18-04996-f006:**
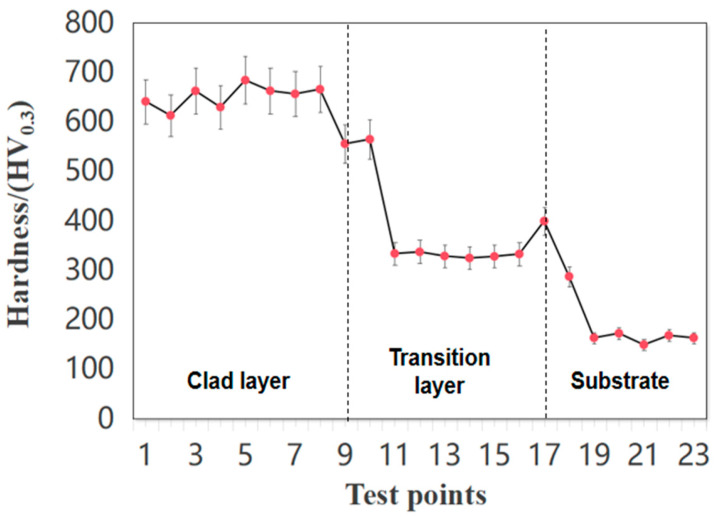
Microhardness distribution of the laminated structure interface.

**Figure 7 materials-18-04996-f007:**
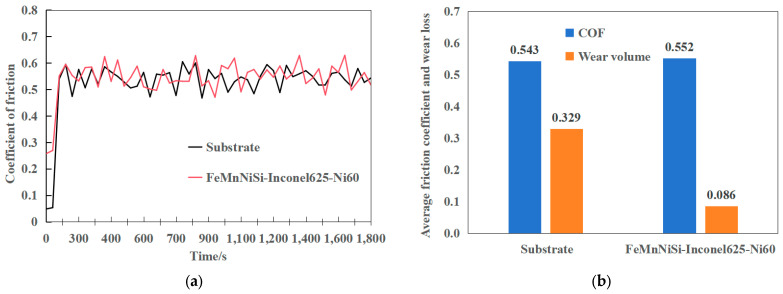
Friction coefficient curve of FeMnNiSi-Inconel625-Ni60 laminated structure cladding layer and substrate. (**a**) Friction coefficient curve. (**b**) Average friction coefficient and wear volume.

**Figure 8 materials-18-04996-f008:**
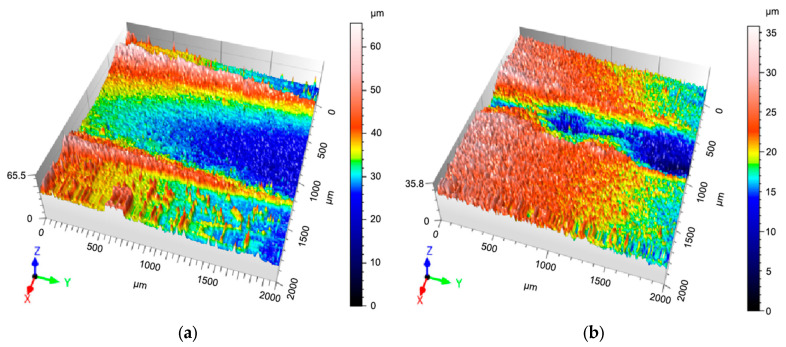
Three-dimensional contours of wear scars on substrate and FeMnNiSi-Inconel625-Ni60 cladding layer. (**a**) Three-dimensional profile of wear scar on substrate. (**b**) Three-dimensional profile of wear scar on FeMnNiSi-Inconel625-Ni60 cladding layer.

**Figure 9 materials-18-04996-f009:**
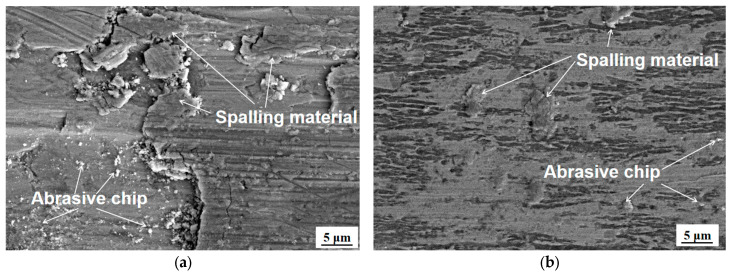
Micromorphology of wear marks on substrate and laminated cladding layer. (**a**) Micromorphology of wear scar on substrate. (**b**) Micromorphology of wear scar on Ni60 cladding layer.

**Figure 10 materials-18-04996-f010:**
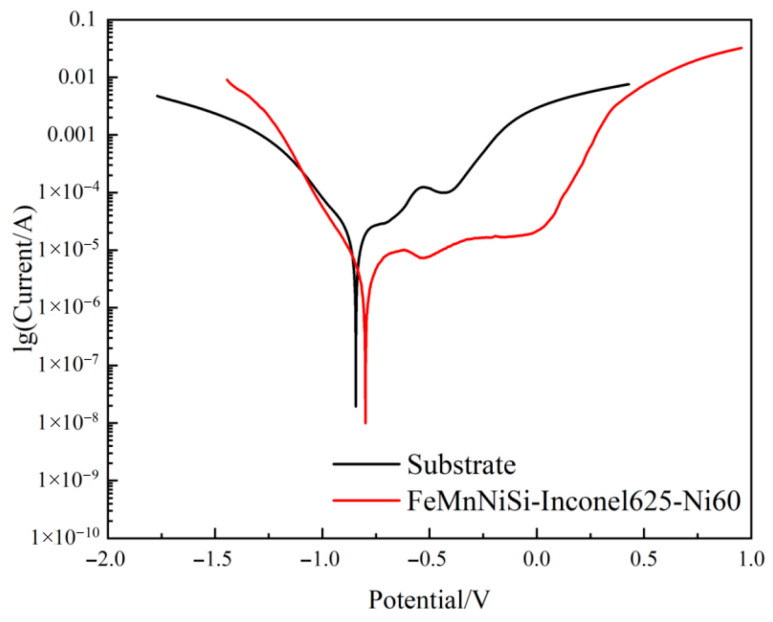
Tafel polarization curves of cladding layer and substrate.

**Figure 11 materials-18-04996-f011:**
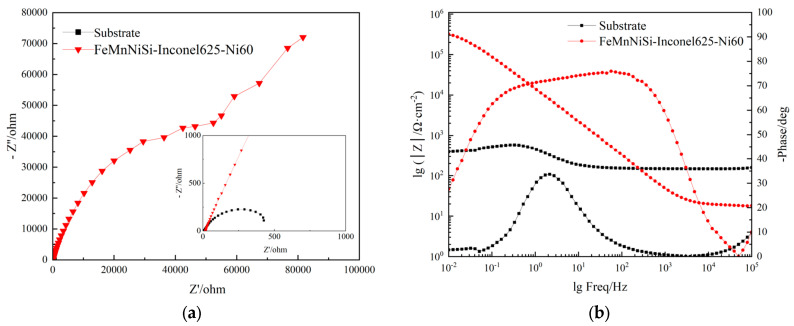
Electrochemical impedance spectroscopy of the laminated structure cladding layer and substrate. (**a**) Nyquist curve of the laminated structure cladding layer and substrate. (**b**) Bode diagram of the laminated structure cladding layer and substrate.

**Figure 12 materials-18-04996-f012:**
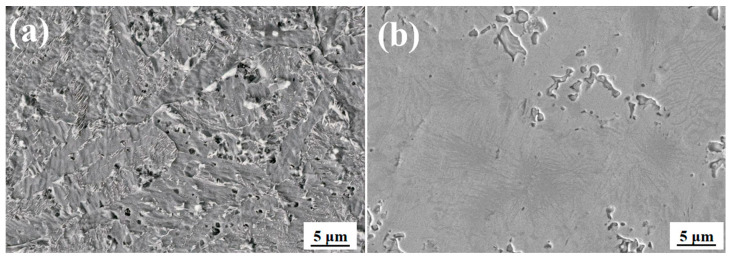
Micromorphology of substrate and laminated structure cladding layer after electrochemical corrosion. (**a**) Micromorphology of substrate. (**b**) Electrochemical corrosion micromorphology of laminated structure cladding layer.

**Table 1 materials-18-04996-t001:** Composition and content of alloy elements in substrate, cladding layer Ni60, and Inconel625 (mass fraction/%).

Material	C	Ni	Cr	Mo	Mn	Nb	B	Si	Fe
FeMnNiSi	0.44	0.20	0.20	—	0.8	—	—	0.30	Bal
Ni60	0.10	60	16	2	—	—	3.5	3.5	Bal
Inconel625	0.10	63	20	8	0.3	3.55	—	0.1	3.0

**Table 2 materials-18-04996-t002:** Wear volume calculation parameters.

Sample	Average Coefficient of Friction	Test Load /N	Sliding Distance /mm	Microhardness /HV_0.3_	Wear Volume /mm^3^
Substrate	0.543	20	5	165.22	0.329
Sandwich structure	0.552	20	5	641.31	0.086

**Table 3 materials-18-04996-t003:** Comparison of friction and wear test data between FeMnNiSi-Inconel625-Ni60 laminated structure and substrate.

Sample	Wear Depth /μm	Wear Width/ μm	Wear Cross-Sectional Area/μm^2^	Wear Volume/ mm^3^	Wear Rate/ mm^3/^(N·m)
Substrate	18.6	780	7806	0.329	3.29 × 10^−2^
FeMnNiSi-Inconel625-Ni60	16.5	460	3883	0.086	0.86 × 10^−2^

## Data Availability

The original contributions presented in this study are included in the article. Further inquiries can be directed to the corresponding author.

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
