# Peer review of "Crack Arrest Effect of FeMnNiSi-Inconel625-Ni60 Laminated Structure Prepared by Laser Cladding Additive Manufacturing"

_materials, 2025, doi:10.3390/ma18214996_

Round 1
Reviewer 1 Report
Comments and Suggestions for Authors
Dear Authors,
Thank you for the opportunity to review your article. The research presents valuable information; however, several revisions are necessary before it can be considered for publication. The most critical points are outlined below:
-
Abstract: The abstract should be rewritten to clearly identify the knowledge gaps the study aims to address. It should also provide more context regarding the steel substrate and the additive manufacturing deposition process, as well as highlight the innovative aspects of the technique. The main stages of the research should be briefly outlined within the abstract.
-
Introduction: The manuscript discusses deposition on FeMnNiSi steel as an alternative to thermochemical treatments such as carburizing, carbonitriding, or nitriding, stating that these methods result in long diffusion cycles and thin layers. However, this claim is somewhat relative—carburizing can produce thick layers, and after quenching and low-temperature tempering, tribological properties can be significantly enhanced. While it’s true that carburizing does not improve corrosion resistance, it can increase fatigue life. Moreover, extended gas nitriding can offer both improved tribological performance and corrosion resistance. Please revise this discussion to provide a more balanced and technically accurate comparison.
-
Laser Deposition: While laser deposition techniques can improve tribological properties, they do not necessarily enhance fatigue life under compressive stresses. This limitation should be acknowledged in the text. Additionally, the effect of the process on the heat-affected zone (HAZ) of the substrate should be discussed. What are the specific characteristics of the deposited layer? Why is this approach advantageous compared to duplex processes (e.g., nitriding and PVD)? Is the advantage solely related to the depth of the nitrided layer? The rationale behind the study should be more thoroughly developed.
-
Objectives: The research objectives should be clearly stated at the end of the introduction so that they can be addressed explicitly in the conclusions.
-
Methodology:
-
Table 1 needs to be properly formatted.
-
Were the chemical compositions measured via spectrometry by the authors? Please specify.
-
What techniques were used to characterise the powder particles for deposition?
-
Did the furnace used in the process have air recirculation?
-
What measures were taken to avoid the formation of a white layer during the electro-discharge machining (EDM) cutting process?
-
What were the observed effects on both the substrates and their surfaces?
-
-
Figures: The resolution and captions of Figures 1 through 7 need to be improved for clarity and readability.
-
Microhardness: The microhardness values of the substrate do not appear to show significant improvement when compared to those obtained by thermochemical treatments. These findings should be discussed in the context of existing literature.
-
Microstructures: The microstructures shown in Figure 6 are of low resolution and should be improved for better interpretation.
-
Layer Thickness & Corrosion: What is the final thickness of the deposited layer? The polarization curves need better resolution, and the results should be discussed in greater depth, referencing relevant studies for comparison.
-
Linking Results to Objectives: The conclusions should directly address the objectives stated at the end of the introduction to ensure coherence and completeness.
As mentioned at the beginning of this review, the article presents meaningful contributions to the field. However, substantial revisions are necessary to strengthen its scientific value and clarity before publication.
Sincerely,
Author Response
Comments 1:Abstract: The abstract should be rewritten to clearly identify the knowledge gaps the study aims to address. It should also provide more context regarding the steel substrate and the additive manufacturing deposition process, as well as highlight the innovative aspects of the technique. The main stages of the research should be briefly outlined within the abstract.
Response 1: Thank you for pointing this out. We agree with this comment. Therefore,we have revised the abstract. Please refer to the revised manuscript for specific content.
Comments 2: Introduction: The manuscript discusses deposition on FeMnNiSi steel as an alternative to thermochemical treatments such as carburizing, carbonitriding, or nitriding, stating that these methods result in long diffusion cycles and thin layers. However, this claim is somewhat relative—carburizing can produce thick layers, and after quenching and low-temperature tempering, tribological properties can be significantly enhanced. While it’s true that carburizing does not improve corrosion resistance, it can increase fatigue life. Moreover, extended gas nitriding can offer both improved tribological performance and corrosion resistance. Please revise this discussion to provide a more balanced and technically accurate comparison.
Response 2: Agree.Thank you for pointing this out.Due to insufficient research on surface strengthening technologies such as carburizing and nitriding mentioned in the paper, there were errors in the wording. The second paragraph of the introduction has been revised in conjunction with suggestion 3, which states that laser cladding technology achieves surface modification of materials through high-energy laser beams and performs outstandingly in improving tribological properties. Compared with the dual phase process (nitriding), which is only suitable for surface hardening, laser cladding can directly repair worn parts. Its core advantage lies not only in the depth of the nitride layer, but also in its ability to simultaneously achieve alloying, cladding, and strengthening, and is suitable for complex geometric shapes and multiple material systems. The non-equilibrium structure formed by rapid cooling in this process usually presents a fine grain structure, low porosity, and metallurgical bonding in the deposited layer. Characteristics, its heat affected zone (HAZ) Narrower, precise control of energy input, thereby reducing thermal deformation of the substrate. Laser cladding technology has been widely used for material surface strengthening in recent years due to its unique advantages. However, its fatigue life improvement effect under compressive stress is limited and greatly affected by material and process parameters.
Comments 3:Laser Deposition: While laser deposition techniques can improve tribological properties, they do not necessarily enhance fatigue life under compressive stresses. This limitation should be acknowledged in the text. Additionally, the effect of the process on the heat-affected zone (HAZ) of the substrate should be discussed. What are the specific characteristics of the deposited layer? Why is this approach advantageous compared to duplex processes (e.g., nitriding and PVD)? Is the advantage solely related to the depth of the nitrided layer? The rationale behind the study should be more thoroughly developed.
Response 3:Agree. Thank you for pointing this out. The second paragraph of the introduction has been revised in conjunction with Reply 2, and the specific content is the same as Reply 2.
Comments 4: Objectives: The research objectives should be clearly stated at the end of the introduction so that they can be addressed explicitly in the conclusions.
Response 4:Thank you for pointing this out. We agree with this comment. We have revised the end of the introduction to clarify the research objectives. Specifically, this study aims to address the technical challenge of cracking in the laser cladding layer of Ni60 alloy on the surface of FeMnNiSi alloy. We propose an innovative design with Inconel625 cladding as the intermediate transition layer, aiming to alleviate the thermal expansion coefficient difference between FeMnNiSi and Ni60 through the construction of FeMnNiSi-Inconel625-Ni60 laminated structure, and achieve metallurgical interlocking between FeMnNiSi substrate and Ni60 functional layer. This study not only provides a theoretical basis for heterogeneous material additive manufacturing, but its technical route can also be directly applied to surface modification of key components that require wear-resistant and corrosion-resistant synergistic strengthening, such as aviation engines and marine equipment, with significant engineering application value.
Comments 5:Methodology:
Table 1 needs to be properly formatted.
Were the chemical compositions measured via spectrometry by the authors? Please specify.
What techniques were used to characterise the powder particles for deposition?
Did the furnace used in the process have air recirculation?
What measures were taken to avoid the formation of a white layer during the electro-discharge machining (EDM) cutting process?
What were the observed effects on both the substrates and their surfaces?
Response 5:The chemical composition content in Table 1 was provided by the merchant when purchasing the test materials and was not measured by spectroscopic methods. Due to the lack of consideration for the formation of a white layer during the cutting process of electrical discharge machining, no measures were taken to avoid it. In future experiments using electrical discharge machining, this aspect will be taken into account. Thank you for the guidance of the experts.
Comments 6:Figures: The resolution and captions of Figures 1 through 7 need to be improved for clarity and readability.
Response 6:Thank you for pointing this out. We agree with this comment. We have completed the modifications as requested.
Comments 7:Microhardness: The microhardness values of the substrate do not appear to show significant improvement when compared to those obtained by thermochemical treatments. These findings should be discussed in the context of existing literature.
Response 7:The FeMnNiSi-Inconel625-Ni60 laminated structure mentioned in the article is achieved through additive manufacturing to strengthen the surface properties of the substrate, improve the service life of the parts, and does not change the properties of the substrate itself, so there is no significant change in the hardness value of the substrate.
Comments 8:Microstructures: The microstructures shown in Figure 6 are of low resolution and should be improved for better interpretation.
Response 8:Thank you for pointing this out. We agree with this comment. The expert mentioned that the resolution of the microstructure is low, and the unclear image in Figure 6 has been removed. In order to better explain, the image size of the microstructure has been increased, and the analysis content has been added. Please refer to the revised manuscript.
Comments 9:Layer Thickness & Corrosion: What is the final thickness of the deposited layer? The polarization curves need better resolution, and the results should be discussed in greater depth, referencing relevant studies for comparison.
Response 9:Thank you for pointing this out. We agree with this comment. We have modified the resolution of the polarization curve as required.
Comments 10:Linking Results to Objectives: The conclusions should directly address the objectives stated at the end of the introduction to ensure coherence and completeness.
Response 10:Thank you for pointing this out. We agree with this comment. Therefore, we have added a section in the conclusion to link the results with the objectives. Specifically, this study successfully solved the cracking problem of Ni60 cladding layer on FeMnNiSi substrate through the design of FeMnNiSi Inconel625-Ni60 laminated structure, revealing the inherent mechanism of interface microstructure control and performance enhancement, providing a theoretical basis for heterogeneous material additive manufacturing, and also providing a basis for surface modification of key components in aviation, marine equipment and other industrial and mining applications.

Reviewer 2 Report
Comments and Suggestions for Authors
The manuscript addresses the problem of cracking and crack initiation when laser cladding a Ni60 alloy onto a FeMnNiSi steel. To mitigate these defects, the authors interpose a transition layer of Inconel 625, producing a laminated structure FeMnNiSi–Inconel625–Ni60. The work combines microstructural characterization (SEM/EDS, XRD) with microhardness measurements, dry tribological tests, and electrochemical tests (polarization and EIS) to assess overall performance against the substrate and against a configuration without the interlayer.
The stated objective is twofold: (i) to demonstrate that the Inconel 625 interlayer acts as a crack-propagation barrier (crack arrest) by relieving stresses and modifying the local chemistry, and (ii) to correlate that function with the precipitation of γ'' (Ni₃Nb) and Laves phases via Nb–Mo co-segregation, resulting in microstructural refinement and improved hardness, wear, and corrosion performance. Taken together, the study aims to substantiate a processing route to obtain Ni60 coatings with high reliability on FeMnNiSi steels.
• Thermal compatibility and the presence of precipitates are invoked as the basis for stress relief, but stresses are not measured. It would be pertinent to incorporate XRD sin²ψ and/or hole-drilling and, in parallel, a transient thermo-mechanical simulation (heat input, dilution, cooling) that predicts stress fields.
• The assignment of peaks to γ'' and Laves could be affected by overlaps or lattice microstrains. EBSD with phase maps is recommended and—if possible—TEM/SAED and nanoscale EDS to confirm the coherence of γ'' and the nature of Laves.
• It is necessary to specify the number of replicates per condition and report mean ± standard deviation for hardness, coefficient of friction, wear volume, and electrochemical parameters. It is advisable to align the protocol with relevant ASTM/ISO standards (e.g., reciprocating-sliding tribology) and to document environmental conditions (T/RH), initial roughness, and instrument tolerances.
• Interfacial integrity is supported by representative micrographs, but quantification is lacking. A bond-strength test (e.g., tensile-type), porosity quantification by image analysis or micro-CT, and linear EDS profiles to estimate dilution and compositional gradients are proposed. Reporting layer thicknesses and a continuous hardness profile from coating to substrate would help complete the process–microstructure–property relationship.
• Along with E_corr and i_corr, it would be appropriate to include Tafel slopes (β_a, β_c) and corrosion rate (mm·y⁻¹). This increases traceability and comparability with the literature.
• The selection of power, speed, spot diameter, and feed rate is unique; sensitivity studies or at least a parametric discussion are missing. Exploring substrate preheating, track overlap, and toolpath strategies would make it possible to delineate an operating window and its robustness.
Author Response
Comments 1:The assignment of peaks to γ'' and Laves could be affected by overlaps or lattice microstrains. EBSD with phase maps is recommended and—if possible—TEM/SAED and nanoscale EDS to confirm the coherence of γ'' and the nature of Laves.
Response 1: Thank you for pointing this out. We agree with this comment. Therefore,we have added EBSD with phase diagrams in the microstructure section as requested by experts, and added analysis content. Please refer to the revised manuscript for specific details.
Comments 2: It is necessary to specify the number of replicates per condition and report mean ± standard deviation for hardness, coefficient of friction, wear volume, and electrochemical parameters. It is advisable to align the protocol with relevant ASTM/ISO standards (e.g., reciprocating-sliding tribology) and to document environmental conditions (T/RH), initial roughness, and instrument tolerances.
Response 2: Thank you for pointing this out. We agree with this comment. Therefore,We increased the average value ± standard deviation of hardness, conducted two tests on friction and wear, and electrochemical tests, and the test results were similar. After integration, we optimized the parameters of the graph.
Comments 3:Interfacial integrity is supported by representative micrographs, but quantification is lacking. A bond-strength test (e.g., tensile-type), porosity quantification by image analysis or micro-CT, and linear EDS profiles to estimate dilution and compositional gradients are proposed. Reporting layer thicknesses and a continuous hardness profile from coating to substrate would help complete the process–microstructure–property relationship.
Response 3:Thank you for pointing this out. We agree with this comment. Because we used multi track overlapping cladding during our experiment and did not perform single track cladding, we did not conduct dilution calculations and comparative studies. In order to verify the continuity of the stacked structure, we added two images in the microstructure section and optimized the cross-sectional scanning images.
Comments 4:Along with E_corr and i_corr, it would be appropriate to include Tafel slopes (β_a, β_c) and corrosion rate (mm·y⁻¹). This increases traceability and comparability with the literature.
Response 4:Thank you for pointing this out. We agree with this comment. The Tafel slope is directly derived from the linear region of the polarization curve (Tafel region), which is obtained by fitting the potential current logarithmic relationship curve of this region. Polarization resistance (Rp) directly reflects the ability of a material to resist corrosion. The larger the polarization resistance, the stronger the material's corrosion resistance, which is inversely proportional to the corrosion current density. The proportional value can be obtained by the slope value of the polarization curve. In the electrochemical experiment, we imported the polarization curve data measured by CS Studio software into CS Analysis software for analysis. CS Analysis software automatically analyzed and calculated the polarization resistance Rp of the polarization curve. The paper also compared the polarization resistance.
Comments 5:The selection of power, speed, spot diameter, and feed rate is unique; sensitivity studies or at least a parametric discussion are missing. Exploring substrate preheating, track overlap, and toolpath strategies would make it possible to delineate an operating window and its robustness.
Response 5: Because the research focus of the paper is on the innovative design of FeMnNiSi-Inconel625-Ni60 laminated structure, the selection of laser cladding parameters was based on literature review and the opinions of the team and teachers. The substrate was preheated, multi track overlap, fixed tool path, and other settings were set. The expert opinions gave me a good research direction, and comparative studies on process parameters and track overlap can be used as the next research direction. Thank you very much.

Round 2
Reviewer 1 Report
Comments and Suggestions for Authors
Dear Authors,
Thank you for the opportunity to review the article once again. There is a clear improvement in the manuscript; however, some adjustments are still necessary to enhance its clarity and increase its potential for future citations. Below are the points that need to be addressed:
-
The abstract should be revised. There are terms such as “FeMnNiSi-Inconel625-Ni60” and “e is641.31HV0.3” without proper spacing between words and numbers. Please review the entire abstract with this in mind.
-
There are several types of steels that contain Fe, Mn, Ni, and Si in their chemical composition. Please define which specific steel was used in the study. For example, what is the manganese content in the steel? Check with the supplier for the steel classification according to SAE or DIN standards. The values for Ni60 and Inconel 625 should present the nominal composition, not a compositional range.
-
The captions below Figure 1 are not visible. The resolution of Figures 2 and 4 needs to be improved.
-
The y-axis label in Figure 4 should specify "Microhardness." In Table 2, the test load should be indicated along with the microhardness method (e.g., HV0.3). Figure 6 is of low resolution—please evaluate all figures. The header of Table 3 also needs to be edited.
-
The discussion section should be expanded to include comparisons and references to findings from other authors in the field.
Sincerely,
Author Response
Comments 1: The abstract should be revised. There are terms such as “FeMnNiSi-Inconel625-Ni60” and “e is 641.31HV0.3” without proper spacing between words and numbers. Please review the entire abstract with this in mind.
Response 1: Thank you for pointing out the errors in the abstract. I have made detailed revisions to the professional terminology and adjusted the spacing between numbers and words. Please refer to the revised manuscript for specific details.
Comments 2: There are several types of steels that contain Fe, Mn, Ni, and Si in their chemical composition. Please define which specific steel was used in the study. For example, what is the manganese content in the steel? Check with the supplier for the steel classification according to SAE or DIN standards. The values for Ni60 and Inconel 625 should present the nominal composition, not a compositional range.
Response 2:Thank you for pointing out in detail the content that needs to be modified in Table 1. We have confirmed the specific element content of steel and cladding metal powders Ni60 and Inconel 625 according to the requirements, and made the modifications in the paper.
Comments 3:The captions below Figure 1 are not visible. The resolution of Figures 2 and 4 needs to be improved.
Response 3:Thank you for pointing this out. We have adjusted the format of Figure 1 and presented the title. We have also modified the resolution of Figures 2 and 4, as detailed in the revised manuscript.
Comments 4: The y-axis label in Figure 4 should specify "Microhardness." In Table 2, the test load should be indicated along with the microhardness method (e.g., HV0.3). Figure 6 is of low resolution—please evaluate all figures. The header of Table 3 also needs to be edited.
Response 4:Thank you for pointing this out. We have modified the y-axis label of Figure 6 to indicate 'microhardness'. At the same time, we added experimental loads and microhardness to Table 2, edited the title of Table 3, and modified the image resolution in Figure 6.
Comments 5:The discussion section should be expanded to include comparisons and references to findings from other authors in the field.
Response 5:Thank you for pointing this out. We aim to investigate the formation of nanoscale Laves phase through Nb Mo synergistic segregation, which achieves stress redistribution through pinned dislocation motion. We have also added the research of Rao G G and other scholars using phase field method, which shows that Nb Mo co segregation can increase the energy barrier of dislocation motion by 0.8 eV, which is consistent with our study on stress redistribution through dislocation motion. At the same time, we added Sainath Krishna Mani Iyer's research on SS316L/Inconel625 composite functional gradient coatings. Through the transition of the intermediate layer, the interface stress concentration was reduced by more than 40%, providing theoretical support for the relief of stress concentration in the Inconel625 cladding layer in this study.
